# TP63 Is Significantly Upregulated in Diabetic Kidney

**DOI:** 10.3390/ijms22084070

**Published:** 2021-04-15

**Authors:** Sitai Liang, Bijaya K. Nayak, Kristine S. Vogel, Samy L. Habib

**Affiliations:** 1Department of Cell Systems and Anatomy, The University of Texas Health Science Center, San Antonio, TX 78229, USA; liangs@uthscsa.edu (S.L.); Nayak@uthscsa.edu (B.K.N.); vogelk@uthscsa.edu (K.S.V.); 2South Texas, Veterans Healthcare System, San Antonio, TX 78229, USA

**Keywords:** IRS-1, TP63, kidney, diabetes

## Abstract

The role of tumor protein 63 (TP63) in regulating insulin receptor substrate 1 (IRS-1) and other downstream signal proteins in diabetes has not been characterized. RNAs extracted from kidneys of diabetic mice (db/db) were sequenced to identify genes that are involved in kidney complications. RNA sequence analysis showed more than 4- to 6-fold increases in TP63 expression in the diabetic mice’s kidneys, compared to wild-type mice at age 10 and 12 months old. In addition, the kidneys from diabetic mice showed significant increases in TP63 mRNA and protein expression compared to WT mice. Mouse proximal tubular cells exposed to high glucose (HG) for 48 h showed significant decreases in IRS-1 expression and increases in TP63, compared to cells grown in normal glucose (NG). When TP63 was downregulated by siRNA, significant increases in IRS-1 and activation of AMP-activated protein kinase (AMPK (p-AMPK-Th^172^)) occurred under NG and HG conditions. Moreover, activation of AMPK by pretreating the cells with AICAR resulted in significant downregulation of TP63 and increased IRS-1 expression. Ad-cDNA-mediated over-expression of tuberin resulted in significantly decreased TP63 levels and upregulation of IRS-1 expression. Furthermore, TP63 knockdown resulted in increased glucose uptake, whereas IRS-1 knockdown resulted in a decrease in the glucose uptake. Altogether, animal and cell culture data showed a potential role of TP63 as a new candidate gene involved in regulating IRS-1 that may be used as a new therapeutic target to prevent kidney complications in diabetes.

## 1. Introduction

Several lines of evidence indicate that insulin resistance is a major issue in treating diabetes and is influenced by multiple physiological and pathological factors [1]. Physiological factors that influence insulin resistance include the glucagon-to-insulin ratio, obesity, and stress. Non-physiological factors that affect insulin resistance including medications such as β-blockers, corticosteroids, oral contraceptives, and pathological conditions, such as Cushing’s disease and inherited metabolic or mitochondrial disorders [1,2,3,4]. Several mechanisms, including defects or abnormalities of the insulin receptor, Glucose transporter type 4 (Glut4), insulin receptor substrate, and phosphorylating phosphatidylinositol-4-5-bisphosphate (PIP-3) kinase, can contribute to insulin resistance and dysfunction of endothelium is another possible mechanism [1,5].

Tumor protein 63 (TP63) acts as a sequence-specific DNA binding transcriptional activator or repressor [6]. TP63 is able to produce several proteins either by splicing or using an alternative promoter, functioning through their transactivation, DNA-binding, or tetramerization domains [7,8,9,10]. TP63 has a major role in the regulation of epithelial proliferation, as well as in the differentiation of epithelia in the mammary glands and lacrimal glands [11,12]. In Tp63-knockout mice, the development of epithelial cells is disrupted and the formation of a p63 deficiency activates widespread cellular senescence with enhanced expression of senescent markers, such as senescence-associated β-galactosidase (SA-β-gal), progressive multifocal leukoencephalopathy (PML), and p16^INK4a^ [13].

TP63 has a similar domain combination with Tumor protein 53 (TP53), which include an N-terminal transactivation domain, a proline-rich sequence, a central DNA-binding domain, and a C-terminal oligomerization domain [8], allowing overexpressed p63 to mimic p53 in terms of biologic activities [6,14,15,16]. TP63 is expressed in a tissue and cell type-specific manner and is present in epithelia, including basal cells in glandular structures, such as the prostate and breast. This is in contrast to the ubiquitous expression of p53 [6,7,17,18,19]. TP63 activates cell cycle genes and enhances cell proliferation in human keratinocytes and MCF7 breast cancer cells [20,21,22]. Recent studies have shown that TP63 is a master regulator of epithelial cells through a combined subset of molecular mechanisms, including cellular energy metabolism and respiration [23]. One of the TP63 isoforms, ΔNp63, is reported to be a major factor in cellular energy metabolism in human keratinocytes by regulating the glycolytic enzyme hexokinase 2, which is the first step of glucose utilization in cells [23]. In addition, strong evidence demonstrates TP63′s important role in maintaining the proliferative potential of epithelial cells [24]. TP63 deficiencies in mice resulted in glucose intolerance, as well as premature aging, obesity, and multiple phenotypes consistent with type 2 diabetes, including glucose intolerance, insulin resistance, and liver steatosis [25,26]. TP63 has an important role in the regulation of energy metabolism by accumulating in response to metabolic stress through the transcriptional activation of Sirtuin 1 (Sirt1), AMP-activated protein kinase (AMPK), and Liver kinase B1 (LKB1), which leads to increased fatty acid synthesis and decreased fatty acid oxidation. The restoration of Sirt1 or AMPK rescued some of the metabolic defects of the TP63^−/−^ mice [25].

Although the involvement of TP63 in epithelial cell proliferation and metabolism is clear, the detailed role of TP63 in regulating insulin receptor substrate 1 (IRS-1) is unknown. In the present study, we investigated the mechanism(s) by which high glucose regulates TP63 and other signal proteins to regulate IRS-1/AMPK/HIF-1/tuberin pathways in proximal tubular renal cells exposed to high glucose and in diabetic mice.

## 2. Results

### 2.1. Physiological Parameters

To test the effects of chronic hyperglycemia on renal physiological parameters, diabetic (db/db) and wild-type (WT) mice were scarified at the age of 10 and 12 months. Kidney hypertrophy measured as the ratio of kidney-weight-to-body-weight is more pronounced in diabetic mice compared to WT mice. Blood glucose levels are significantly higher in diabetic mice compared to WT mice. Diabetic mice have a higher albumin/creatinine ratio, and protein in urine from diabetic mice is also significantly increased compared to WT mice at both ages (Table 1). Altogether, these data show that physiological parameters of kidney function in diabetic mice are more severely affected by chronic hyperglycemia than in WT mice.

### 2.2. TP63 RNA Expression Is Significantly Increased in Diabetic Mice

RNA sequence analysis showed more than 4- to 6-fold reads in TP63 expression in the kidneys of diabetic mice, compared to wild-type mice, at ages 10 and 12 months, respectively (Figure 1A,B). To confirm the up-regulation of TP63 at the transcriptional level, TP63 mRNA expression was measured by quantitative reverse transcription poly chain reaction (qRTPCR) analysis. TP63 mRNA levels are significantly upregulated in diabetic mice, compared to the WT, at the 10- and 12-months (Figure 1C,D). These data confirmed that TP63 expression is significantly increased in diabetic mice compared with WT mice.

### 2.3. TP63 Is Significantly Upregulated in the Kidneys of Diabetic Mice

To confirm the upregulation of TP63 at the translational level in diabetic mice at ages of 10 and 12 months, kidney cortices from WT and db/db animals were homogenized, and the lysates were subjected to Western blot analysis. The data in Figure 2 show a significant increase in TP63 expression in diabetic mice compared to WT mice and are consistent with the increased mRNA expression levels (Figure 2A,B) at both ages. Immunofluorescent staining for TP63 was performed on kidney sections and confirmed that TP63 is significantly upregulated in diabetic mice (Figure 2C–F). Note that the majority of TP63 staining was detected within tubular cells (marked with arrows). These data are consistent with the results from Western blot in WT and diabetic mice at both ages and confirm that TP63 is significantly upregulated in diabetic mice and that the TP63 up-regulation is associated with chronic hyperglycemia.

### 2.4. Downregulation of IRS-1 Is Correlated with Upregulation of TP63 in Diabetic Mice

Kidney cortices from WT and db/db mice were homogenized, and the lysates were subjected to Western blot analysis. The downregulation of IRS-1 expression is correlated with the upregulation of TP63 in diabetic mice, as compared with WT mice, at both 10 and 12 months of age (Figure 2G,I). Next, we measured the expression levels of genes downstream from IRS-1 in kidney tissue homogenates from WT and diabetic mice, using Western blot analysis (Figure 2G,I,K). In contrast, the expression of HIF1a, also downstream from IRS-1, is significantly increased in diabetic mice (Figure 2H,J,K). These data suggest a key role for TP63 in the regulation of the IRS-1/p-AMPK/tuberin/HIF1a pathway in the diabetic kidney.

### 2.5. Downregulation of TP63 Resulted in Activation of IRS-1 Pathway

Our immunohistochemical data indicated that TP63 levels are increased in the kidney tubular cells of diabetic mice. To identify the mechanisms through which TP63 may regulate IRS-1, murine proximal tubular epithelial (MCT) cells were transfected with siRNA-TP63 under NG or HG conditions. The downregulation of TP63 by siRNA-TP63 resulted in significant increases in IRS-1, p-AMPK, and tuberin protein levels and a decrease in HIF1a protein expression under NG and HG conditions (Figure 3A–C), compared to non-transfected cells by Western blot analysis. These data suggest that TP63 might be a target of IRS-1/p-AMPK/tuberin/HIF1a pathways.

### 2.6. Downregulation of IRS-1 Increases TP63 Protein Expression

MCT cells transfected with siRNA to knockdown IRS-1 showed increases in p-AMPK expression (Figure 3D,E). In addition, the downregulation of IRS-1 resulted in increased expression of HIF1a, whereas tuberin expression is significantly decreased under both NG and HG conditions (Figure 3F). 

### 2.7. Glucose Uptake Controls by TP63 and IRS-1

The inhibition of glucose uptake is a measure of insulin resistance. IRS-1 facilitates glucose uptake and promotes insulin sensitivity. To examine the role of TP63 and IRS-1 in glucose uptake, MCT cells were transfected with siRNA-TP63 or siRNA-IRS-1. The knockdown of TP63 resulted in an increase, whereas the knockdown of IRS-1 resulted in a decrease in glucose uptake (Figure 3G) compared to non-transfected cells (control). Cytochalasin B, a glucose transport inhibitor, completely inhibited glucose uptake. These data are consistent with a mechanism whereby TP63 may contribute to insulin resistance by decreasing glucose uptake in cells.

### 2.8. AMPK Alters TP63 Levels and Controls IRS-1 Expression

To further investigate the mechanisms whereby TP63 regulates downstream target genes of IRS-1, we measured the levels of other target genes of TP63. In one set of experiments, MCT cells were treated with a potent AMPK activator, AICAR (2 mM), and in another set, AMPK levels were downregulated through DN-AMPK for 48hrs. Cells treated with AICAR exhibited a significant decrease in TP63 and an increase in IRS-1 protein expression. In contrast, cells transfected with DN-AMPK exhibited a significant increase in TP63 and a decrease in IRS-1 protein expression (Figure 4A,D). Additionally, p-AMPK and tuberin expression levels are significantly increased following the AICAR treatment and significantly decreased following the DN-AMPK treatment (Figure 4B,C,E,F). In contrast, HIF1a expression decreases following AICAR treatment and increases following treatment of MCT cells with DN-AMPK (Figure 4C,F). Altogether, these data suggest that AMPK blocks TP63 to altered IRS1 protein expression.

### 2.9. Tuberin Controls TP63 Protein Expression

To further investigate the mechanisms whereby TP63 regulates downstream signals of IRS-1 through other target genes, MCT cells were infected with AdTSC2 to overexpress tuberin. Cell lysates were then used to perform Western blot analyses. With overexpression of TSC2 (tuberin), TP63 levels are significantly decreased in both NG and HG conditions compared to non-infected cells (Figure 4G). In contrast, IRS-1 levels are significantly increased when tuberin is overexpressed in both NG and HG conditions, with the increase being more pronounced under HG conditions (Figure 4G). In response to TSC2 overexpression, HIF1a expression is significantly decreased under NG and HG conditions compared to non-infected cells (Figure 4G). Knockdown of tuberin using siRNA resulted in increased TP63 and HIF1a expression and decreased IRS-1 protein expression (Figure 3H). These data suggest that tuberin blocks TP63 expression, which in turn alters IRS-1 protein expression. 

### 2.10. Downregulation of IRS-1 Significantly Increases TP63 in Fresh Primary Proximal Tubular Cells

To confirm the link between IRS-1 and TP63, primary proximal tubular cells isolated from WT mice were transfected with either siRNA-HIF1a or siRNA against IRS-1, and the cell lysates were used to perform Western blot analyses. The data showed that the downregulation of IRS-1 with siRNA significantly increases the expression of TP63 and decreases p-AMPK levels (Figure 5A). In addition, the downregulation of IRS-1 results in a significant decrease in tuberin and an increase in HIF1a expression (Figure 5C). On the other hand, the downregulation of HIF1a by siRNA-HIF1a significantly decreases the expression of TP63, with a more pronounced decrease under HG conditions. p-AMPK protein levels are increased when HIF1a is downregulated. (Figure 5B). These data are consistent with the results obtained from MCT cells and suggest the TP63 is upregulated under high glucose conditions and may lead to developing diabetes complications.

## 3. Discussion

This study provides clear evidence that TP63 is a key factor in regulating the IRS-1 pathway in kidney tubular cells through its ability to negatively regulate IRS-1, which has a well-characterized role in transducing the insulin signal. We used several different approaches to demonstrate the regulation of TP63 through IRS-1 in the context of the kidney. First, RNA sequence analysis showed that TP63, a new candidate gene, is highly expressed and associated with significant decreases in IRS-1 levels in diabetic mice’s kidneys. Second, the downregulation of TP63 significantly increases IRS-1 protein expression and alters glucose uptake in MCT cells. Third, the activation of AMPK by AICAR or inactivation of AMPK by DN-AMPK alters TP63 and IRS-1 expression in MCT cells. Fourth, adenovirus-mediated overexpression of tuberin, and the downregulation of tuberin by siRNA, alters both TP63 and IRS-1 expression in MCT cells. These results were also confirmed in fresh primary proximal tubular renal mouse cells to provide strong evidence for a new role for TP63 in regulating the IRS1/MPK/tuberin/HIF1a pathways, thus contributing to the development of diabetes complications.

Insulin resistance is a major issue in both basic research and clinical treatment of diabetes. In most situations, insulin resistance is manifested at the cellular level as related to the insulin-signaling system, especially at the level of post-receptor defects [1]. Unfortunately to date, the mechanism of insulin resistance remains elusive, although the data from animal experiments related to insulin-receptor defects are promising [1]. In previous studies, several mechanisms of insulin resistance were proposed, including the defects of insulin receptor (IR), the downregulation or genetic changes in IR, and the defects of IRS-1 or PIP-3 kinase pathways. TP63 is reported to be involved in regulating cell metabolism and energy production [23,24,25,26]. One of the TP63 isoforms, ΔNp63, is reported to be a major factor in cellular energy metabolism in human keratinocytes through its regulation of the glycolytic enzyme hexokinase 2, which is the first step of glucose utilization in cells [23]. TP63 also regulates energy metabolism by accumulating in response to metabolic stress and transcriptionally activating Sirt1, AMPK, and Liver kinase B1 (LKB1); thus, leading to increased fatty acid synthesis and decreased fatty acid oxidation. Restoration of Sirt1 or AMPK rescues some of the metabolic defects of the TP63^−/−^ mice [25].

Our data show that TP63 is significantly upregulated at both mRNA and protein levels in diabetic mice compared to WT mice, and this increase is associated with the downregulation of IRS-1. These data suggest that the downregulation of IRS-1 significantly increases TP63 protein expression in diabetic mice. TP63 immunostaining was localized within tubular cells in kidney sections, with increased levels observed in diabetic mice compared to WT mice, suggesting that TP63 is upregulating under hyperglycemia and specific to tubular cells. In proximal tubular cells, the downregulation of TP63 resulted in significant increases in IRS-1and HIF1a and concomitant decreases in tuberin expression. Moreover, experiments with freshly isolated primary proximal tubular cells confirmed that TP63 negatively regulates IRS-1. In contrast to our findings with kidney cells, TP63 positively regulates IRS-1 in squamous cell carcinoma, giving cancer cells a growth advantage [27,28,29]. 

Hyperglycemia resulted in the downregulation of IRS-1 and induction of TP63, which may lead to diabetic complications. Thus, decreasing TP63 expression could restore the expression of IRS-1. Additionally, the activation of AMPK by AICAR decreased the expression of TP63 and increased the expression of IRS-1, whereas the inactivation of AMPK by DN-AMPK increased TP63 expression and decreased IRS-1 expression. Moreover, overexpression of tuberin resulted in decreased TP63 and increased IRS-1 expression. Glucose uptake was increased by the knockdown of TP63 and decreased by the knockdown of IRS-1. Inhibition of glucose uptake is a measure of insulin resistance (Figure 6). Altogether, these data support our hypothesis that TP63 regulates IRS-1 in diabetes, thus contributing to the development of diabetic complications. In summary, these data showed TP63′s potential role as a new candidate gene that may be involved in insulin resistance in diabetes through regulating glucose uptake.

## 4. Materials and Methods

### 4.1. Animals

Male db/db mice (Strain BKC.Cg.m +/+ Leprdb/J) and wild-type (WT) mice were purchased from Jackson Laboratory. Food and water ad libitum were allowed prior to and during the experiments to all animals. WT and db/db mice were euthanized at 10 and 12 months old, and their kidneys were dissected for biochemical analysis. All animals used in this study were approved (10087x, July, 2016) by the Institutional Animal Aare and Use Committee (IACAC) at the University of Texas Health Science Center, San Antonio, TX, USA.

### 4.2. Physiological Parameters of Mice

Blood glucose was measured by a glucometer in overnight fasting mice. Twenty-four hours of urine was collected from each mouse individually in the metabolic cages. Creatinine concentration in urine and serum was measured using the ELISA kit from BioAssay System (Creatinine Assay Kit, Hayward, CA, USA). Albumin concentration was measured in urine by the ELISA kit from the Immunology Consultants Laboratory (Portland, OR, USA). The total protein concentration was measured in urine by the Bradford assay using bovine serum albumin as a standard [30]. The body and kidney weights of all mice were recorded on the day of sacrificing.

### 4.3. RNA Sequence Data Analysis

Reads were aligned to the transcriptome reference FASTA file, which was downloaded from the National Center for Biotechnology Information (NCBI) archive (build.37.2). Data quality check, preprocessing, and cleaning were done using our in-house pipeline as previously described [31].

### 4.4. qRT-PCR of TP63

RNA was extracted from the kidney using the RNeasy Mini Kit from Qiagen, (Germantown, MD, USA). RNA was quantified by spectrophotometer at 260 nm. Real-time PCR was performed as previously described [32] using the primers of TP63.

### 4.5. Cell Culture

The murine proximal tubular epithelial (MCT) cells were grown in DMEM containing 10% fetal bovine serum, 5-mmol/L glucose, 100-units/mL penicillin, 100 μg/mL streptomycin, and 2 mmol/L glutamine. Overnight serum-free DMEM medium was used overnight before experiments. The cells were grown in a humidified atmosphere of 5% CO_2_ at 37 °C.

### 4.6. Isolation and Culture of Fresh Mouse Proximal Tubular Epithelial Renal Cells

Primary cells were isolated from the kidneys of wild-type (WT) mice and cultured as previously described [33] with minor modifications. The cells were incubated at 37 °C in humidified 5% CO_2_ in air, and the medium was changed every 6 days until the cells reached 60% confluence. 

### 4.7. Cells Treatment

MCT cells were grown in 5 mM normal glucose (NG) or treated with 25 mM high glucose (HG). Cells were pretreated with AICAR (2 mM) prior to exposure to HG for 48 h as previously described [33]. AICAR was obtained from Cayman Chemical (Ann Arbor, MI, USA). The cell lysates were used for Western blot analysis.

### 4.8. Downregulation of AMPK, HIF-1, IRS-1 and TP63

MCT cells were transfected at 50–60% confluence with a recombinant plasmid expressing DN-AMPK prior to exposure to high glucose. The plasmid containing AMPK carrying K45R mutation of the α1-subunit (pCAGGS) was transfected into the cells using lipofectamine and A-plus reagent (Invitrogen) as previously described [34]. Knockdown of HIF-1, IRS-1, and TP63 in MCT cells was achieved using siRNA against HIF-1, IRS-1, and TP63. All siRNAs were purchased from Santa Cruz Biotech (Santa Cruz, CA, USA). Transfection of siRNA was performed by using the oligofectamine reagent following the company standard protocol (Invitrogen, Carlsbad, CA, USA). The cells were then incubated for 48 h at 37 °C in a humidified atmosphere of 5% CO_2_. The cell lysates were used for Western blot analyses. The cells were treated with HG (25 mM) 24 h before harvesting for Western blot analysis. 

### 4.9. Overexpression and Downregulation of Tuberin

MCT cells were grown to 40–50% confluence, made quiescent by serum deprivation for 24 h, and then infected with adenovirus 6.01 carrying the cDNA of the TSC2 gene (Clontech Laboratories, San Francisco, CA, USA) as previously described [35]. On the other hand, the knockdown of tuberin in MCT cells was achieved using siRNA against tuberin. siRNA-tuberin was purchased from Santa Cruz Biotech (Santa Cruz, CA, USA). Transfection of siRNA was performed by using the oligofectamine reagent following the company standard protocol (Invitrogen, Carlsbad, CA, USA). The cells were then incubated for 48 h at 37 °C in a humidified atmosphere of 5% CO_2_. The cells were treated with HG (25 mM) 24h before harvesting for Western blot analysis.

### 4.10. Western Blot

The cell lysates homogenates or kidney cortex were prepared for Western blot analysis as previously described [36]. Protein concentrations were determined using the Bradford assay reagent [37]. Western blot analysis was performed as described previously [38]. Tuberin, HIF1a, p-AMPK, AMPK, tubulin, and IRS-1 antibodies were purchased from Cell Signaling (Boston, MA, USA). The GADPH antibody was obtained from Santa Cruz Biotechnology (Santa Cruz, CA, USA). The TP63 antibody was purchased from Thermo Scientific (Waltham, MA, USA). Protein expression was identified byusing an enhanced chemiluminescence kit (Amersham, Piscataway, NJ, USA). The expression of each protein was quantified by densitometry using the National Institutes of Health image J software (Bethesda, MD, USA) and normalized to the loading control.

### 4.11. Immunofluorescence Staining of TP63

TP63 immunostaining was performed in mice kidney sections as described previously [38]. Fluorescein isothiocyanate (FITC) green signals for TP63 were detected using a filter with an excitation range of 450 nm to 490 nm using a filter with the excitation at 535 nm. The Nikon Research microscope equipped for epifluorescence with excitation and band-pass filters was used to view and photograph kidney sections. To demonstrate staining specificity, control kidney sections were stained without a primary antibody.

### 4.12. Glucose Uptake Assay

The glucose uptake assay was performed as previously described [39]. MCT Cells were seeded into a 48-well plate. The next day, the cells were transfected with siRNA-TP63 or siRNA-RS-1 using oligofectamine reagent (Invitrogen). The cells were serum-starved for 4 h. Afterwards, 2.5 µL of transport solution (1 volume of 3H 2DG + 20 volumes of 1 mM 2-DG) was added to each well. The glucose uptake assay was performed 48 h after the transfection of the cells and measured as count per minute (CPM). Glucose transport was stopped by adding 2 volumes of 50 mM glucose after 30 min. The cells were washed 3 times in cold phosphate buffer saline (PBS). The cells were lysed by adding 0.6 m Glul of 0.1 M NaOH to each well and incubated for 15 min. The cell lysate (0.5 mL) was taken for 3H counting using a scintillation counter. Non-transfected cells were used as control to measure the basal glucose uptake. In one experimental control group, the cells were treated with 100 µM cytochalasin B (glucose transport inhibitor) before adding the transport solution.

### 4.13. Statistics

Data are presented as the mean ± standard error. Statistical differences were determined using ANOVA followed by Student Dunnett’s (Exp. vs. Control) test using 1 trial analysis. *p*-values less than 0.01 were considered statistically significant.

## 5. Conclusions

Our data provide clear evidence that TP63 is a key factor in regulating IRS-1 and other downstream target genes (Figure 6). We identified TP63 was significantly upregulated when IRS-1 decreased under hyperglycemia in diabetic mice and in renal cells exposed to high glucose. These data demonstrate TP63′s potential role as a new candidate gene involved in regulating IRS-1 to regulate glucose uptake, which may be involved in insulin resistance in diabetes.

## Figures and Tables

**Figure 1 ijms-22-04070-f001:**
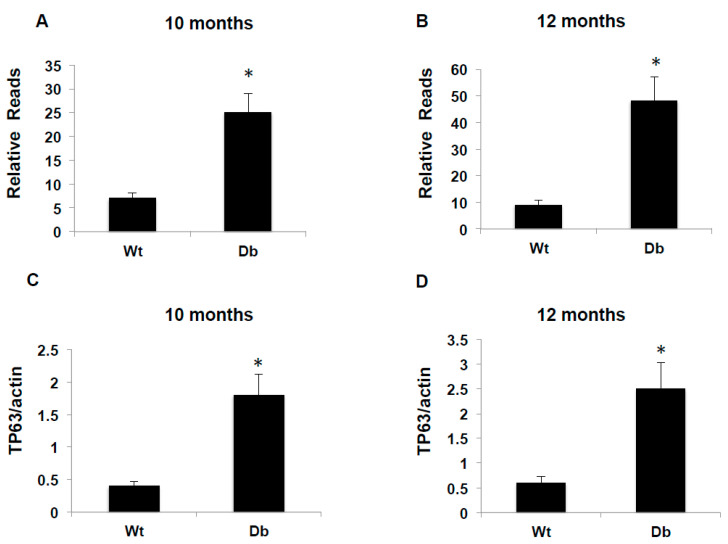
A Significant increase in RNA reads and tumor protein 63 (TP63) mRNA in diabetic mice. Kidney tissues from WT and diabetic mice were homogenized, and the total RNA was extracted. RNA reads were measured by RNA sequence in 10-month-old (**A**) and 12-month-old (**B**) mice. (**C**,**D**) mRNA of TP63 quantitation was performed by qRTPCR showed higher expression in diabetic mice compared to WT mice. A significant difference from WT mice is indicated by * *p* < 0.01.

**Figure 2 ijms-22-04070-f002:**
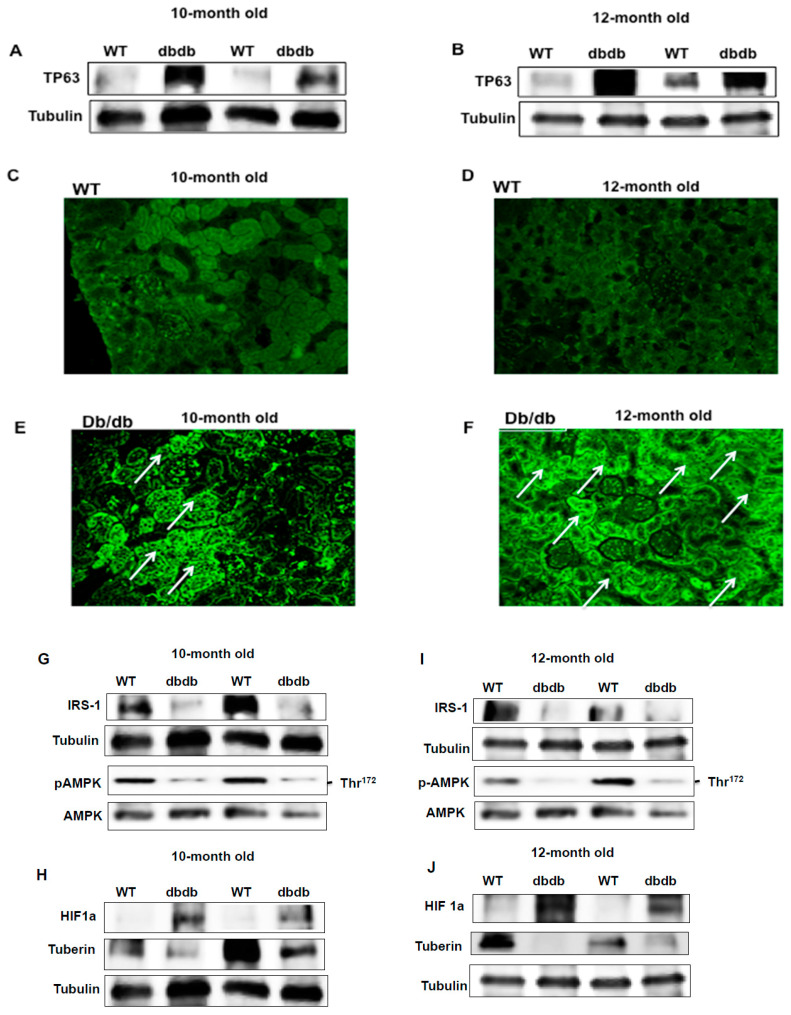
Increased expression of TP63 is associated with a decrease in insulin receptor substrate 1 (IRS-1), phospho-AMP-activated protein kinase (p-AMPK), and tuberin in kidneys of diabetic mice. (**A**,**B**) Data from diabetic kidneys showed an increase in TP63 expression and a decrease in IRS-1 expression compared to WT mice at 10- and 12-months-old. (**C**–**F**) Immunofluorescence staining confirmed increased expression of TP63 in diabetic mice of 10- and 12-month-old age compared to WT. Note that the majority of TP63 staining was localized within tubular cells (marked with arrows). (**G**,**I**) Increased expression of TP63 is associated with a decrease in p-AMPK expression in 10- and 12-month-old diabetic mice, compared to WT mice. (**H**,**J**) Increased expression of TP63 is also associated with a higher expression of HIF1a and a lower expression of tuberin in diabetic mice compared to WT mice at both ages. (**K**) Histogram presented quantification Western blot data of each protein normalized by tubulin and p-AMPK normalized by AMPK of 12 months old mice (*n* = 4) groups. A significant difference from the WT group is indicated by * *p* < 0.01.

**Figure 3 ijms-22-04070-f003:**
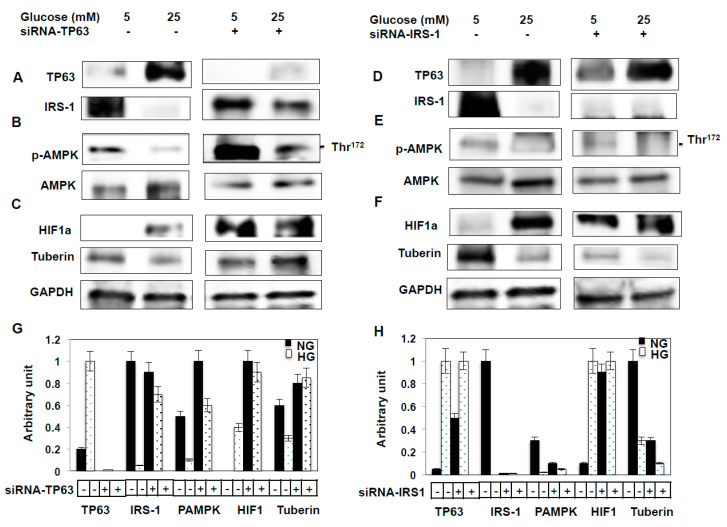
The downregulation of TP63 increases IRS-1, resulting in increased glucose uptake. Mouse tubular (MCT) cells were transfected with siRNA against TP63 or IRS-1 in normal glucose (NG) and high glucose (HG) conditions. (**A**) TP63 is downregulated significantly by TP63 siRNA in both NG and HG. The downregulation of TP63 increased the expression of IRS-1 significantly. (**B**) Both p-AMPK and AMPK were increased significantly by downregulating TP63 in both NG and HG compared to non-transfected cells. (**C**) HIF1a is significantly increased in HG-treated cells, while the downregulation of TP63 resulted in decreased expression of HIF1a. (**D**) The downregulation of IRS-1 by siRNA-IRS-1 in NG and HG resulted in a significant increase in TP63 expression under NG and HG conditions. (**E**) The downregulation of IRS-1 decreased p-AMPK significantly compared to non-transfected cells. (**F**) HIF1a was significantly increased with the downregulation of IRS-1, while tuberin expression was decreased under both NG and HG conditions. (**G**,**H**) Histogram of average expression of each protein presented as an arbitrary unit was calculated from each protein and normalized by Glyceraldehyde 3-phosphate dehydrogenase (GAPDH) from 2 Western blots. (**I**) Glucose uptake (CPM) is increased by inhibiting TP63 and decreased by inhibiting IRS-1. Cells were transfected either with siRNA-TP63 or siRNA-IRS-1, resulting in an increase and decrease in glucose uptake compared to the non-transfected cells (control), respectively. Cells treated with glucose transport inhibitor cytochalasin B resulted in complete inhibition of glucose transport. Data presented as means ± SE (*n* = 6). A significant difference from non-transfected cells is indicated by * *p* < 0.01 and from cells transfected with siRNA-IRS-1 or treated with cytochalasin B groups by ** *p* < 0.001.

**Figure 4 ijms-22-04070-f004:**
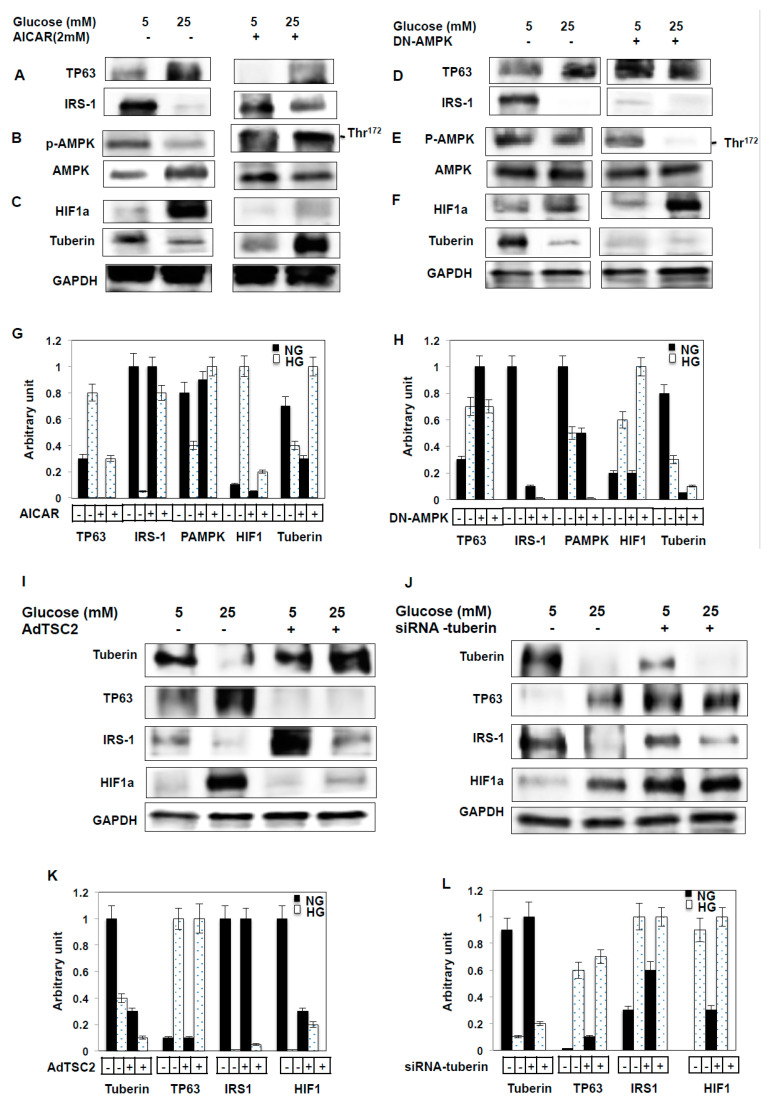
Activation of AMPK by AICAR decreases TP63 and increases IRS-1, and over-expression of tuberin resulted in decreased TP63 expression. MCT cells were treated with AICAR (2mM) or transfected with DN-AMPK. (**A**) AICAR treatment abolished TP63 expression significantly in NG and HG, while IRS-1 expression was decreased in NG and increased in HG. (**B**) AICAR treatment increased the expression of both p-AMPK and AMPK significantly. (**C**) AICAR treatment downregulated HIF1a expression in NG, and the expression of tuberin was downregulated significantly in both NG and HG. (**D**) DN-AMPK transfection significantly downregulated TP63 expression and increased IRS-1 expression. (**E**) DN-AMPK treatment downregulated expression of p-AMPK. (**F**) DN-AMPK transfection increased the expression of HIF1a greatly and significantly downregulated the expression of tuberin under both NG and HG conditions. (**G**,**H**) Histogram of the average expression of each protein presented as an arbitrary unit was calculated from each protein and normalized by GAPDH from 2 Western blots. (**I**,**J**) The overexpression of tuberin resulted in significant decreases in TP63 and greatly increased IRS-1 expression compared to non-infected cells under NG and HG conditions. In contrast, HIF1a is significantly downregulated with the overexpression of tuberin and increased with the siRNA-tuberin treated cells compared to non-infected cells under NG and HG conditions. (**K**,**L**) The average expression of each protein presented as an arbitrary unit was calculated from each protein and normalized by GAPDH from 2 Western blots.

**Figure 5 ijms-22-04070-f005:**
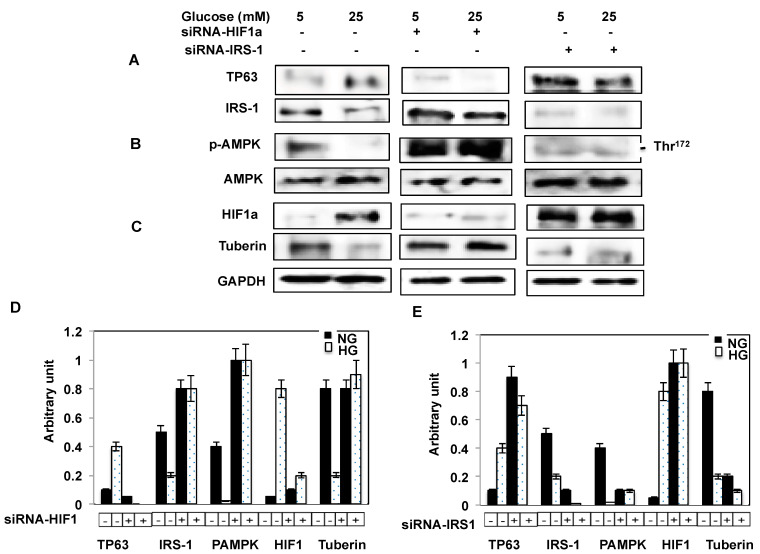
The downregulation of HIF1a and IRS-1 altered TP63 expression in fresh primary proximal tubular renal cells. Primary cells grown in NG or HG condition were infected with siRNA-HIF1a or transfected with siRNA against IRS-1. (**A**)The expression of TP63 is significantly decreased with the downregulation of HIF1a and greatly increased with the downregulation of IRS-1. (**B**) The expression of p-AMPK is significantly decreased with the downregulation of IRS-1 and increased with the downregulation of HIF-1a. (**C**) The downregulation of HIF1a increased tuberin expression, while the downregulation of IRS-1 decreased tuberin protein expression. (**D**,**E**) Histogram of the average expression of each protein presented as an arbitrary unit was calculated from each protein and normalized by GAPDH from 2 Western blots.

**Figure 6 ijms-22-04070-f006:**
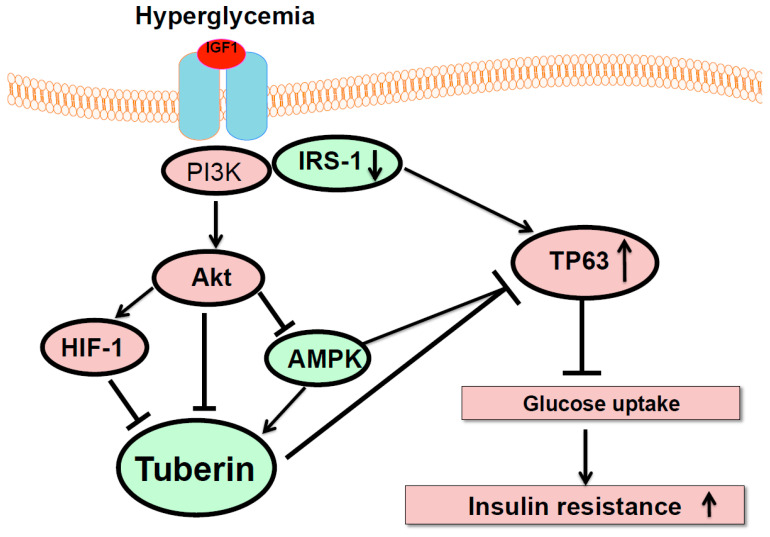
The proposed model of TP63’s role in insulin resistance. Hyperglycemia decreases IRS-1 to activate Akt/HIF-1 and downregulate tuberin/AMPK. The downregulation of IRS-1 increases TP63 under high glucose to decrease glucose uptake, suggesting that TP63 may be involved in insulin resistance in diabetes.

**Table 1 ijms-22-04070-t001:** Physiological parameters of wild-type (WT) and diabetic mice (db/db) at 10 and 12 months old. Blood glucose level is much higher in diabetic mice than in WT mice, as expected. Both body weight and kidney/body weight ratio are higher in diabetic mice than in WT mice. Albumin/Creatinine ratio is higher in diabetic mice compared with WT mice. Diabetic mice have more proteinuria compared with WT mice at both 10- and 12-months-old. A significant difference from WT mice is indicated by * *p* < 0.01.

	WT 10 Months	Dbdb 10 Months	WT 12 Months	Dbdb 12 Months
**Body weight (g)**	30.64 ± 7.0	36.19 ± 6.5	30.11 ± 2.6	37.03 ± 6.9
**Kidney/body weight (g/g)**	0.926 ± 0.2	1.0 ± 0.2 *	0.96 ± 0.2	1.12 ± 0.2 *
**Glucose levels (mg/dL)**	119.4 ± 22.9	379.5 ± 65.5 *	90.8 ± 18.6	391.0 ± 90.9 *
**Albumin/Creatinine ratio (μg/mg)**	109.8 ± 20.6	530.6 ± 109 *	117.5 ± 19.7	650.4 ± 187 *
**Proteinurea (mg/24h)**	21.75 ± 8.0	35.44 ± 12.3 *	26.36 ± 6.7	42.58 ± 7.6 *

## Data Availability

Not applicable.

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
