# Peer review of "TP63 Is Significantly Upregulated in Diabetic Kidney"

_ijms, 2021, doi:10.3390/ijms22084070_

Round 1

Reviewer 1 Report

The study titled “TP63 is a New Candidate Gene Involved in Insulin Resistance” aimed to investigate the role of TP63 in the insulin signaling pathway. Although the study has some merits, there are serious issues that need to be considered.

Major issues:

Why 10- and 12-months aged mice are chosen for the study? what is the rationale? Why tubular renal cells were studied for the mechanism of action?

The proposed figure for the role of TP63 in insulin resistance (Figure 6) lacks several important components and needs to be revised. The current version shows that IRS-1/PI3K/Akt activation leads to HIF-1 and TP63 overexpression and inhibited glucose uptake which is not correct.

Western blot data is only qualitative, and no quantification has been done on the density of protein bands. Authors are suggested to quantitatively determine the protein abundance of the studied candidates.

For glucose uptake, no insulin has been added to the media, did the authors measured basal glucose uptake? How the insulin signaling is studied under normal glucose and high glucose conditions without insulin addition to the media?

Minor issues:

The introduction does not give enough background for the study. Authors keep using TP63 or p63 interchangeably which makes it hard to understand the story. Please keep that clear

Mixed English writing styles are apparent throughout the manuscript. Several typos and extra wordings and grammatical flaws are present.

Figure 2 panels G-J don’t have group indicators

Author Response

Major issues:

  1. Why 10- and 12-months aged mice are chosen for the study? what is the rationale? Why tubular renal cells were studied for the mechanism of action?

We chose age of 10 and 12 months of diabetic mice to test the effect of chronic hyperglycemia on altering the physiological parameters as well as on regulation of cell signals proteins including TP63. We have included the rationale in page 5.

We used renal tubular cells to study the mechanism of action because we found that majority of tubular cells stained with TP63 (see Fig. 2 E&F)

  1. The proposed figure for the role of TP63 in insulin resistance (Figure 6) lacks several important components and needs to be revised. The current version shows that IRS-1/PI3K/Akt activation leads to HIF-1 and TP63 overexpression and inhibited glucose uptake, which is not correct.

We agree with the reviewer that there regulation of insulin resistance has several important components but we proposed the signals that we studied in our manuscript. We revised the scheme to simply present the effect of major molecules that involved in regulating TP63 in our study (See Fig. 6).

  1. Western blot data is only qualitative, and no quantification has been done on the density of protein bands. Authors are suggested to quantitatively determine the protein abundance of the studied candidates.

We have included the quantitation of western bots.

  1. For glucose uptake, no insulin has been added to the media, did the authors measured basal glucose uptake? How the insulin signaling is studied under normal glucose and high glucose conditions without insulin addition to the media?

Yes, we measured the basal levels of glucose uptake under normal condition (column 1 in Fig. 3G). We didn’t use insulin as it is known to increase glucose uptake and our study was only focused on the effect of TP63 and IRS-1 on glucose. All singling related to regulation of IRS-1/TP63 and other proteins were performed under normal and high glucose in cell culture as well as under normal glucose and hyperglycemia in mice (Fig. 1-5)

Minor issues:

  1. The introduction does not give enough background for the study. Authors keep using TP63 or p63 interchangeably which makes it hard to understand the story. Please keep that clear

We have revised the introduction and corrected the typo of TP63 and p63 in the revised manuscript.

  1. Mixed English writing styles are apparent throughout the manuscript. Several typos and extra wordings and grammatical flaws are present.

We have revised the English in the whole manuscript.

  1. Figure 2 panels G-J don’t have group indicators

We have added the group ages each panel from G-J in Fig. 2.

Reviewer 2 Report

This is interesting and novel study that examines the role of transcription factor, TP63, in IRS1-mediated signaling and insulin resistance in kidney cells, which are relevant to the origin and etiology of diabetic nephropathy. Using combination of biochemical and microscopy approaches the authors provided a clear and conclusive evidence that TP63 is a key factor in regulating insulin resistance by negatively regulating IRS-1. The is a well-executed study, supported by a number of quantitative and qualitative experiments/data. I have a few, minor comments and suggestions for the authors that may further improve quality and visibility of this otherwise novel and rigorous study.

- Table 1: Please include glycosylated hemoglobin levels, if available.

- Fig. 2 G-J: It would be helpful if treatments are listed for each of these blots, as done for Fig. A-F. Also, densitometric analysis would be helpful here to indicate level of proteins upregulation or downregulation, as done in Fig. 3G. Further, the authors may consider to use graphical symbols (i.e. arrows or arrowheads) to indicate TP63-positive tubular cells. In addition to this, the authors should discuss here a notable difference in TP63 staining pattern between control and Db/Db mice.

- In 2.4 section, it may be more accurate to use term " correlates" rather than " associates", as no causal relationship was investigated in this part of the studies. These are predominantly comparative studies.

- Fig, 3G: What is n value (# of repetitions) for WB experiments? This information should be added to the fig. legend together with other statistical information.

- For Fig 4, a significant change in protein levels were reported.  It would be very helpful if the authors can provide densitometric analysis/data and provide p values, as they have done in Fig. 3 G.

- Finally, the manuscript should be carefully edited for English and clarity. Few sentences contained repeated statements of experimental findings whereas some listed undefined acronyms (i.e. TP63 vs. p63).

Author Response

  1. This is interesting and novel study that examines the role of transcription factor, TP63, in IRS1-mediated signaling and insulin resistance in kidney cells, which are relevant to the origin and etiology of diabetic nephropathy. Using combination of biochemical and microscopy approaches the authors provided a clear and conclusive evidence that TP63 is a key factor in regulating insulin resistance by negatively regulating IRS-1. This a well-executed study, supported by a number of quantitative and qualitative experiments/data.

We thank the reviewer for positive and encouragement comments.

  1. I have a few, minor comments and suggestions for the authors that may further improve quality and visibility of this otherwise novel and rigorous study. Table 1: Please include glycosylated hemoglobin levels, if available.

We regret to say that don’t have data of glycosylated hemoglobin levels

  1. Fig. 2 G-J: It would be helpful if treatments are listed for each of these blots, as done for Fig. A-F. Also, densitometric analysis would be helpful here to indicate level of proteins upregulation or downregulation, as done in Fig. 3G. Further, the authors may consider to use graphical symbols (i.e. arrows or arrowheads) to indicate TP63-positive tubular cells. In addition to this, the authors should discuss here a notable difference in TP63 staining pattern between control and Db/Db mice.

We have included the age of each group of mice in each panel of Fig. 2. We have included histogram to show upregulation and downregulation of proteins. We have included arrows to mark the staining of tubular cells in kidney sections of diabetic mice (see Fig. 2E&F). We also discussed the pattern of TP63 between control and diabetic mice in discussion section.

  1. In 2.4 section, it may be more accurate to use term " correlates" rather than " associates", as no causal relationship was investigated in this part of the studies. These are predominantly comparative studies.

We have corrected the word to correlate in 2.4 section.

  1. Fig, 3G: What is n value (# of repetitions) for experiments? This information should be added to the fig. legend together with other statistical information.

We have included number for Fig. 3G and also the statistical data.

  1. For Fig 4, a significant change in protein levels were reported.  It would be very helpful if the authors can provide densitometric analysis/data and provide p values, as they have done in Fig. 3 G.

We have included quantitation of all Western blots data.

  1. Finally, the manuscript should be carefully edited for English and clarity. Few sentences contained repeated statements of experimental findings whereas some listed undefined acronyms (i.e. TP63 vs. p63).

We have revised the English in the whole manuscript. We have corrected the typo of TP63 and p63.

Round 2

Reviewer 1 Report

The revised manuscript has improved considerably. There are still a few more issues that need further clarification.

The quantification of WB data has been done adequately in Fig 1 making it more convenient for readers to interpret the data themselves without referring to the text. Authors are suggested to adopt a similar approach for subsequent figures or think about a better way to summarize their data to show the key message of each figure easier. The current presentation of tables for the average protein bands (normalized to GAPDH) is not formative, nor the sole presentation of lots of W/B images...Using n=2 experiments is not usually enough for experimental design, and presentation of the variability of the data is helpful to see if the data is robust and how much error is associated with each quantification.

There are also some other minor issues:

Line 32: change “including” to “include”

Acronyms should be spelled out at their first use in the introduction e.g. Glut4, PIP3, TP63, etc.

Table 1. keep significant digits consistent throughout the table

Fig 2. Panel K, the p-AMPK needs to be normalized to AMPK, not tubulin. The new analysis is unlikely to change the conclusion, but for the sake of accuracy and quality of the paper, authors are suggested to change the analysis and modify the figure.

Author Response

All changes in the revised manuscript made in blue.

1. The quantification of WB data has been done adequately in Fig 1 making it more convenient for readers to interpret the data themselves without referring to the text. Authors are suggested to adopt a similar approach for subsequent figures or think about a better way to summarize their data to show the key message of each figure easier. The current presentation of tables for the average protein bands (normalized to GAPDH) is not formative, nor the sole presentation of lots of W/B images...Using n=2 experiments is not usually enough for experimental design, and presentation of the variability of the data is helpful to see if the data is robust and how much error is associated with each quantification.

We deleted the tables with average of protein bands and used these data to generate histograms and to show the difference in each protein expression under different conditions.

2. There are also some other minor issues:

Line 32: change “including” to “include”

Corrected

3. Acronyms should be spelled out at their first use in the introduction e.g. Glut4, PIP3, TP63, etc.

All acronyms spelled out.

4. Table 1. keep significant digits consistent throughout the table

Corrected

5. Fig 2. Panel K, the p-AMPK needs to be normalized to AMPK, not tubulin. The new analysis is unlikely to change the conclusion, but for the sake of accuracy and quality of the paper, authors are suggested to change the analysis and modify the figure.

P-AMPK normalized to AMPK.

Round 3

Reviewer 1 Report

The revised manuscript has been considerably improved.